# Novel Compounds in the Treatment of Schizophrenia—A Selective Review

**DOI:** 10.3390/brainsci13081193

**Published:** 2023-08-11

**Authors:** Evangelia Maria Tsapakis, Kalliopi Diakaki, Apostolos Miliaras, Konstantinos N. Fountoulakis

**Affiliations:** 13rd Department of Academic Psychiatry, AHEPA General Hospital, 546 36 Thessaloniki, Greece; 2Department of Psychiatry, Academic General Hospital, 711 10 Heraklion, Greece

**Keywords:** schizophrenia, psychosis, aetiopathogenesis, novel antipsychotics, olanzapine/samidorphan, lumateperone, brilaroxazine, Kar-XT, emraclidine, ulotaront, pimavanserin, sodium benzoate, luvadaxistat, iclepertin

## Abstract

Schizophrenia is a chronic neuropsychiatric syndrome that significantly impacts daily function and quality of life. All of the available guidelines suggest a combined treatment approach with pharmacologic agents and psychological interventions. However, one in three patients is a non-responder, the effect on negative and cognitive symptoms is limited, and many drug-related adverse effects complicate clinical management. As a result, discovering novel drugs for schizophrenia presents a significant challenge for psychopharmacology. This selective review of the literature aims to outline the current knowledge on the aetiopathogenesis of schizophrenia and to present the recently approved and newly discovered pharmacological substances in treating schizophrenia. We discuss ten novel drugs, three of which have been approved by the FDA (Olanzapine/Samidorphan, Lumateperone, and Pimavanserin). The rest are under clinical trial investigation (Brilaroxazine, Xanomeline/Trospium, Emraclidine, Ulotaront, Sodium Benzoate, Luvadaxistat, and Iclepertin). However, additional basic and clinical research is required not only to improve our understanding of the neurobiology and the potential novel targets in the treatment of schizophrenia, but also to establish more effective therapeutical interventions for the syndrome, including the attenuation of negative and cognitive symptoms and avoiding dopamine blockade-related adverse effects.

## 1. Introduction

Schizophrenia is a neuropsychiatric disorder, most likely due to neurodevelopmental reasons arising from abnormalities in different circuits and neurotransmitter systems. This developmental process includes uncontrolled synaptic pruning, deriving from gene–environment interaction [1]. Numerous studies suggest that a synaptic pathology, rather than a loss of neurons, is at the forefront [2]. In addition to the reduction in the number of synapses, multiple findings suggest that both the synapse structure and function are affected. For example, a decrease in the cortical pyramidal neuron dendritic spine density and dendritic branching, especially in layer III, has been a consistent finding in schizophrenia [3]. In addition, most loci from GWAS studies on the genetics of schizophrenia encode structurally and functionally critical synaptic proteins [4].

The dopamine (DA) hypothesis of psychosis is one of the most enduring ideas in psychopharmacology. Undeniably, one central dysfunction stems from dopaminergic abnormalities, but in a different way than initially thought (updated DA hypothesis—the mesostriatal hyperdopaminergia) [5]. Moreover, DA is not the only neurotransmitter linked to psychosis, as many other neurotransmitter systems are dysregulated. At present, numerous alternative neurotransmitter theories are complementary rather than contradictory to the DA hypothesis. Moreover, diverse hypotheses are interconnected rather than conflicting. For example, the loss of dendritic spines in cortical layer III is linked to the DA hypothesis. The loss of dendritic spines and NMDAR hypofunction in inhibitory interneurons cause an excitatory/inhibitory (E/I) imbalance and a decrease in γ-aminobutyric acid (GABA)ergic transmission, resulting in cortical E/I imbalance. This imbalance leads to the overstimulation of the subcortical areas and the overactivation of the mesolimbic pathway, inducing positive symptoms [6].

Increasing evidence implicates glutamate, cholinergic, and serotonin neuronal networks, trace amines, and their TAAR receptors. Focusing on these alternative neurotransmitters/receptors is critical to fully understand the underlying pathophysiology of the psychoses, in addition to developing drugs with novel mechanisms of action other than DA receptor antagonism; the latter results from the necessity for safer, more tolerable, and more effective medications [7]. Improved efficacy includes the control of DA receptor blocking-resistant positive symptoms and the unmet need to alleviate negative and cognitive symptoms.

In this review, we aim to present an update on the current understanding of the aetiopathogenesis of schizophrenia, along with the most promising novel agents in the treatment of this disorder, namely olanzapine/samidorphan, lumateperone, brilaroxazine, xanomeline/trospium, emraclidine, ulotaront, pimavanserin, sodium benzoate, other DAAO inhibitors, D-serine, and iclepertin.

## 2. Methods

MEDLINE was searched for articles written in the English language from inception to 31/7/23 that matched any combination of the following keywords: ((olanzapine/samidorphan[tiab] OR lumateperone[tiab] OR (RP5063[tiab] OR brilaroxazine[tiab]) OR (xanomeline[tiab] OR Kar-XT[tiab]) OR (emraclidine[tiab] OR CVL-231[tiab]) OR (ulotaront[tiab] OR SEP363856[tiab]) OR pimavanserin[tiab] OR sodium benzoate[MeSH] OR DAAO inhibitor*[tiab] OR D-serine[tiab] OR (iclepertin[tiab] OR OR voltage-gated sodium channel blocker[tiab]) AND (schizophr*[MeSH] OR schiz*[tiab]) AND (clinical trial[tiab] OR trial[tiab])). Using the same keywords, the search was repeated for each drug separately in the www.clinicaltrials.gov database. We considered all clinical trials evaluating patients strictly with schizophrenia-type psychosis. We also checked the references in publications found to acquire any additional data, as well as the official sites of the pharmaceutical companies developing the novel compounds in question.

## 3. Results

Our MEDLINE search yielded 102 articles in total, 90 of them concerning humans, and 87 in the English language. Of these, 34 were excluded as irrelevant. In MEDLINE, there were four (4) articles on the olanzapine/samidorphan combination, eight (8) on lumateperone, one (1) on brilaroxazine (RP5063), six (6) on xanomeline/tropsium, one (1) on emraclidine, four (4) on ulotaront, four (4) on pimavanserin, nine (9) on sodium benzoate, one (1) on the DAAO inhibitor luvadaxistat, and fifteen (15) on D-serine. Our www.clinicaltrials.gov search yielded ongoing trials for each compound under investigation (Table 1). The eligibility assessment of the studies was implemented by two authors separately, KD and AM, and the final decision for inclusion was discussed with EMT.

### 3.1. Neurotransmitter System Aetiopathogenesis of Schizophrenia

#### 3.1.1. Dopamine

##### The Classic Dopamine Hypothesis

The classic concept of striatal functioning proposes that the dorsal striatum regulates motor movement, and the ventral striatum regulates emotions via the nigrostriatal and the mesolimbic pathway, respectively [8]. However, in untreated schizophrenia, dopaminergic activity is unaltered in the ventral striatum, or even reduced, and related to negative symptoms [9], but in the associative striatum—an intermediate structure receiving input from the substantia nigra rather than the ventral tegmental area (VTA)—dopamine is overactive. Consequently, rather than separate nigrostriatal and mesolimbic projections, a mesostriatal pathway arising from the VTA–substantia nigra complex and projecting to the associative striatum may present an improved concept [9].

##### New Concepts concerning DA Dysfunction

According to the dopamine synthesizing capacity (DSC) theory, dopaminergic dysfunction predates the subsequent onset of frank psychotic illness in people with symptoms that are truly prodromal to a psychotic disorder [10]. In refractory schizophrenia, however, patients do not exhibit the elevation in DSC classically associated with the disorder, possibly due to a different underlying pathophysiology or a differential effect of antipsychotic treatment [11].

At present, it is widely accepted that the DA receptors in the prefrontal cortex (PFC) maintain a balance of excitation/inhibition (E/I) by modulating glutamate neurotransmission and GABAergic interneuron function [12]. In schizophrenia, this DA-induced cortical fine-tuning is underactive. Moreover, aberrant pruning in the PFC induces the loss of dendritic spines in layer III, possibly leading to further cortical E/I imbalance [13,14].

The neurodevelopmental theory of schizophrenia suggests that DA has pruning properties, mostly notable during the early stages of neuronal development. Excessive pruning is considered the cause of decreased brain volume, which is the most consistent brain abnormality in schizophrenia. Evidence indicates that DA induces a decrease in neurite extensions, mainly via the activation of D1 receptors [15], and neurons developed from the stem cells of patients with schizophrenia present shorter neuronal extensions [16]. The presence of shorter neuronal extensions possibly indicates that a subset of patients with schizophrenia may be more susceptible to DA’s pruning effects during neuronal development [17,18].

#### 3.1.2. Serotonin

Hallucinogens, including the naturally occurring mescaline and psilocybin and the synthetic lysergic acid diethylamide (LSD), act via the serotonin 5-HT2A receptors (5-HT2AR) and enhance glutamatergic transmission to induce their profound changes in human consciousness, emotions, and cognition. In the prefrontal cortex, 5-HT2AR stimulation increases the release of glutamate [19]. The serotonin (5-HT) system inhibits the dopaminergic function at the VTA in the midbrain and at the dopaminergic terminals in the PFC [20]. Considering clozapine’s antipsychotic effect, a DA-serotonin receptor ligand, it has been suggested that, in schizophrenia, the stress-induced overload of serotonin from the dorsal raphe nucleus (DRN) disturbs the activity of cortical neurons and that chronic widespread stress-induced serotonergic overdrive in the cerebral cortex, especially in the anterior cingulate cortex (ACC) and dorsolateral frontal cortex, underlies the development of the disorder [21]. Moreover, hyperactivity/imbalance of serotonin activity, particularly at the serotonin receptors regulating glutamate release, can result in psychosis [22].

Antagonism of the 5-HT2AR receptors on glutamatergic neurons is a common feature of second-generation, or atypical, antipsychotics. 5-HT2ARs are excitatory, leading to excitatory glutamate release at downstream targets, and thus to the hyperactivation of the mesostriatal pathway and hypoactivation of the mesocortical pathway. The serotonin hyperfunction hypothesis of psychosis suggests that psychosis may be caused by an imbalance in excitatory 5-HT2A receptor stimulation on the glutamate pyramidal neurons, which directly innervate mesostriatal DA neurons and visual cortex neurons, inducing delusions and hallucinations. This theory explains the psychotic symptoms in PD dementia and the psychotic effect of 5-HT2A agonists such as psilocybin and LSD. Moreover, it indicates the promising role of pure 5-HT2A antagonists in psychosis [23].

The inhibitory 5-HT1A receptors are located on the glutamate neurons projecting to the VTA or the substantia nigra (SN). When activated by serotonin or a 5-HT1A agonist, they inhibit glutamatergic neurons to enhance mesocortical and nigrostriatal DA pathway neurotransmission [24]. The 5-HT1A receptors are also located on GABA interneurons in the PFC and indirectly regulate the release of norepinephrine (NE), DA, and acetylcholine (ACh). Serotonin or an external agonist binding at these receptors could reduce the GABA output and, in turn, disinhibit NE, DA, or ACh release, thus improving negative and cognitive symptoms. This explains the putative role of 5-HT1A agonists and partial agonists in schizophrenia [25].

Beyond the 5-HT1A and 5HT2A receptors, other serotonin receptors, including the 5-HT2C, 5-HT3, 5-HT6, and 5-HT7 receptors, continue to represent promising drug targets for the discovery of novel multi-receptor antipsychotic agents aiming to address cognitive and negative symptoms [26].

Moreover, serotonin antagonists improve the extrapyramidal side-effects of antipsychotics [27,28], and the antagonism of 5-HT7 receptors may be partially responsible for the antidepressant and procognitive activity of amisulpride [29]. In addition, 5-HT2A, 5-HT2C, and 5-HT3 receptor gene polymorphisms have been associated with antipsychotic-induced weight gain [30,31].

#### 3.1.3. Glutamate

Glutamate is the main and most common excitatory neurotransmitter in the mammalian brain [32]. Dysfunction of the corticolimbic glutamatergic neurotransmission plays a critical role in schizophrenia [33,34]. There are five glutamate pathways: (i) the cortical brainstem glutamate projection descending from cortical pyramidal neurons in the PFC to brainstem neurotransmitter centers (raphe, locus coeruleus, ventral tegmental area, substantia nigra) to regulate neurotransmitter release; (ii) the cortico-striatal glutamate pathway descending from the PFC to the striatum and the nucleus accumbens; (iii) the thalamocortical glutamate pathway ascending from the thalamus to innervate pyramidal neurons in the cortex; (iv) the corticothalamic glutamate pathway descending from the PFC to the thalamus; and (v) the cortico-cortical pathways, composed of intracortical pyramidal neurons which can communicate with each other via glutamate neurotransmission [35]. Glutamatergic neurons represent the primary excitatory afferent and efferent systems innervating the cortex, limbic regions (e.g., hippocampus and amygdala), and striatum [36,37] and orchestrate intricate interplays amongst neuronal networks (e.g., glutamatergic, GABAergic, dopaminergic, serotonergic neurotransmission). Dysfunction in one of those neuronal networks alters the E/I balance [38]. The glutamate theory of psychosis proposes that the NMDA glutamate receptor is hypofunctional at critical synapses in the PFC [39].

Disturbances in glutamatergic neurotransmission may influence synaptic plasticity and cortical microcircuitry, especially NMDA receptor functioning [40]. NMDA receptors belong to ligand-gated ion channels and are important for excitatory neurotransmission, excitotoxicity, and plasticity [41]. The NMDA-receptor hypofunction state is thought to lead to morphological and structural brain changes, resulting in the development of psychosis [42].

The key NMDA receptors are in the indirect cortico-cortical glutamate pathways [35]. Through a cascade of events, one pyramidal neuron inhibits another indirectly via GABA interneurons. More specifically, the first pyramidal neuron excites a GABA interneuron via the NMDA receptors located upon the latter. The GABA interneuron inhibits the second pyramidal neuron, and so on. The glutamate hypothesis suggests that psychosis may be caused by the dysfunction of glutamate synapses at the PFC GABA interneurons. NMDA dysfunction in the PFC leads to the loss of function of the inhibitory GABA interneurons. Thus, the glutamate neurons become ‘disinhibited’, and therefore hyperactive, leading to the downstream effects on the different DA pathways [43]. Moreover, GABAergic neurons in the VTA, which comprise approximately 30% of the total cells in the VTA, play a dual role. VTA GABA neurons provide both local inhibition on VTA DA neurons and long-range inhibition on distal brain regions [44].

Disturbances in glutamate signaling may be an attractive drug target for schizophrenia due to its key role in the aetiopathogenesis of the disorder regarding cognitive impairment and negative symptoms [45]. Antipsychotics may influence glutamate transmission by affecting glutamate release, interacting with glutamatergic receptors, or changing the density or subunit composition of glutamatergic receptors [46]. In this context, dopamine–glutamate interactions occur intraneuronally and intrasynaptically. Antipsychotics also influence glutamate transmission by acting on serotonin receptors [47,48].

However, classical agonists at the NMDA did not prove to be useful as the excessive stimulation of NMDA receptors results in excitotoxicity and neuron damage. The glycine modulatory binding site on the NMDA receptor is considered a more promising target, and direct or indirect glycine modulatory site modulators, including glycine, D-cycloserine, D-serine, glycine transporter 1 (GlyT1) inhibitors, and D-amino acid oxidase (DAAO) inhibitors, have been actively researched in clinical trials and, as discussed in more detail below, show promise in the treatment of schizophrenia following the glutamatergic hypothesis of schizophrenia [48].

#### 3.1.4. Acetylcholine

We have now discussed the dopaminergic, serotoninergic, and glutamatergic hypotheses of schizophrenia. There is also growing evidence that inhibitory GABA signaling is dysregulated in schizophrenia, particularly in the cortex. Dopaminergic, GABAergic, and glutamatergic signaling are all modulated by the cholinergic system; therefore, the modulation of ACh receptors represents an exciting new target. Muscarinic ACh receptors (mAChR) appear to be suitable for interventions. The M1, M4, and M5 mAChR subtypes can all modulate schizophrenia-related circuitry and are intriguing targets for novel schizophrenia treatments [43].

Using traditional therapies that lack specificity among mAChR subtypes requires drug tailoring to hit a therapeutic window between beneficial effects (e.g., pro-cognitive efficacy) and classical cholinergic side-effects. These side-effects are mediated by peripherally expressed M2 and M3 receptors, making the selective targeting of M1, M4, and M5 therapeutically desirable. ACh binds on the mAChRs’ orthosteric binding pocket, which is a specific area on each subtype of mAChR, making it challenging to create molecules that specifically target this site with high subtype selectivity. However, some agonists have been developed that, while lacking complete subtype-selectivity, exhibit preferential activation for certain subtypes over others. Allosteric agents have proved easier in targeting binding pockets outside the orthosteric binding pocket binding ACh. In particular, selective positive allosteric modulators (PAMs) have significantly advanced subtype-selectivity [49].

Substantial evidence supports that the cholinergic system robustly modulates striatal DA signaling through the activation of both mAChRs and nicotinic ACh receptors (nAChRs) via cholinergic projections from the brainstem to the midbrain, where they stimulate DA neurons in SN and the VTA [50].

The M4 receptor is among the best-studied muscarinic targets for psychosis. In the striatum, the M4 subtype is primarily expressed postsynaptically on direct pathway spiny projection neurons (GABAergic receptors of the ‘go-signal’), on cholinergic interneurons, where it acts as an autoreceptor, and on glutamatergic inputs, where it acts as a heteroreceptor [43].

The M4 receptors on cholinergic interneurons and spiny projection neurons modulate DA signaling through different mechanisms. The activation of M4 autoreceptors on cholinergic interneurons reduces ACh release. Typically, ACh induces striatal DA release by activating nAchR on the DA axonal projection, arriving from the VTA/SN complex. Consequently, an M4-mediated decrease in ACh secretion leads to a reduction in DA output. Furthermore, the activation of the M4 receptors on spiny projection neurons leads to an endocannabinoid-dependent blockade of DA release independent of nAChRs. Beyond its role at the striatum, the M4 receptor regulates the hippocampal circuitry involved in cognitive processes. Recent findings suggest that M4 activation (by PAMs) alleviates cognitive deficits and positive symptom reduction [51].

The M1 receptor is another well-studied muscarinic receptor. There is robust evidence that M1 activation leads to pro-cognitive effects and alleviates negative-type symptoms [52]. This effect is mainly mediated via the cortical and hippocampal M1 receptors. Both regions are highly involved in memory processing and express M1 receptors known to modulate neuronal function. Data suggests that M1 receptor expression is reduced in approximately 25% of schizophrenia patients [53].

Moreover, deficits in M1 receptor expression are brain region-specific. In schizophrenia, M1 receptor-expressing neurons are reduced in the cortex but not the thalamus or the hippocampus. As for the M1-selective compounds, pure PAMs seem superior in their pro-cognitive efficacy and exhibit fewer adverse side effects. Although M1 is primarily considered a target for enhancing cognition, there are indications that M1 activation is associated with an antipsychotic effect. Such implications derive from animal studies where M1 PAMs have demonstrated efficacy in positive symptoms [54]. The norclozapine issue also raises the possibility of an M1-mediated antipsychotic effect. Norclozapine is clozapine’s primary metabolite, an agonist at mAChRs with robust activity at the M1 receptor. No other currently used antipsychotics or metabolites are known to act as mAChR agonists. It was assumed that this agonist activity of norclozapine may be one of the unique properties distinguishing clozapine from other atypical antipsychotics in terms of its efficacy in cognitive deficits and resistant positive symptoms [55].

The M5 receptors need to be better studied, and data are scarce. However, they modulate DA signaling via actions at the DA terminals and DA cell bodies, making them an interesting target for the future [56]. The advent of highly M5-selective compounds with acceptable central nervous system (CNS) penetration brings new knowledge on this receptor and its potential to regulate substance abuse disorders. The discovery of new, improved molecules selectively targeting M5 could help to elucidate the potential efficacy of M5-selective NAMs and PAMs in positive and negative symptoms of schizophrenia [43].

#### 3.1.5. Trace Amines and the TAAR Receptors

Trace amines are endogenous substances found in trace levels in the body. They are formed from amino acids inside CNS during the synthesis of monoamines when certain enzymatic steps are omitted. They are commonly found in foodstuffs and can be produced and degraded by microbiota. These molecules do not act as typical neurotransmitters; namely, they are not released upon nerve firing. There are five main human trace amines: β-Phenylethylamine, p-Tyramine, Tryptamine, p-Octopamine, and p-Synephrine [57].

Trace amine-associated receptors (TAARs) are G-protein receptors. There are six isoforms in humans, and TAAR1 is the most studied. Except for TAAR1, all other TAAR subtypes are expressed only in olfactory neurons. TAAR1 are present in multiple CNS areas, including several monoamine brainstem centers and projection areas. They are also expressed in the periphery, where they may have a role in nutrient-induced hormone secretion [58]. Cross-talk between TAAR1 and principal neurotransmitter systems has been demonstrated so that TAAR1 seem to act as a rheostat of DA, Glu, and 5-HT [59]. As a result, trace amines acting upon these receptors are thought to retain neurotransmission within the “physiological” limits of function, in line with the TAAR1 localization in critical regions of the relative neurotransmitter pathways.

TAAR1 are predominantly located intracellularly. The putative mechanism of DA rheostasis is the following. When the TAARs are occupied by an endogenous or an exogenous agonist, the derivative complex is translocated to the cell surface, where it couples with D2. This heterodimerization directs the second-messenger pathway to move preferentially towards the inhibitory G (Gi) protein signal transduction cascade, inhibiting the synthesis and release of DA, rather than towards the β-arrestin-2 excitatory pathway (production of GSK-3 and overstimulation). Presynaptically, the amplification of the Gi pathway leads to the inhibition of DA release, while postsynaptically, the amplification of the Gi pathway can lead to the reduced production of GSK-3 and, therefore, to less stimulation. Consequently, drugs that target TAAR1 constitute a promising field of indirect dopamine modulation [60,61].

## 4. New Agents for the Treatment of Schizophrenia

### 4.1. Olanzapine/Samidorphan

The Olanzapine/Samidorphan (OLZ/SAM) combination was approved by the FDA for the treatment of adults with schizophrenia or bipolar disorder type I in June 2021. Strictly speaking, this is a novel combination of a well-established antipsychotic, olanzapine, with samidorphan. This treatment can be used for maintenance monotherapy or the acute treatment of psychotic, manic, or mixed episodes. Olanzapine, acting primarily via the DA and 5-HT receptors, is a second-generation (atypical) antipsychotic (SGA) approved for treating schizophrenia and bipolar disorder type I. Samidorphan is a 3-carboxamido-4-hydroxynaltrexone, acting as an opioid antagonist, preferentially on the *μ*-opioid receptor.

Enlighten-1 was a 4-week, phase III, randomized, double-blind, placebo- and olanzapine-controlled study conducted in patients with schizophrenia in acute exacerbation [62]. It aimed to evaluate the antipsychotic efficacy and safety of OLZ/SAM. In the 4-week study, with 352 participants, OLZ/SAM treatment showed significant improvement compared to the placebo. Measured by the PANSS score change, OLZ/SAM was as effective as olanzapine versus placebo. The combination was well-tolerated and had a similar safety profile to olanzapine. The most common adverse events for both were weight gain, somnolence, and dry mouth.

Enlighten-2 evaluated the weight gain profile of OLZ/SAM versus olanzapine over 6 months in 561 patients with stable schizophrenia [63]. Compared to olanzapine, the combination of OLZ/SAM resulted in less weight gain and a smaller increase in waist circumference. As expected, the efficacy of the combination was similar to that of olanzapine alone, but the group taking olanzapine had a higher weight gain at 6 months. Specifically, the olanzapine group had a higher number of patients who gained 10% or over of their baseline body weight (*p* = 0.003). The combination was well-tolerated, with the most common adverse events being weight gain, somnolence, and dry mouth.

A systematic review of the literature in 2022 [64] highlighted the positive impact of samidorphan supplementation to olanzapine, on its overall tolerability, and especially on olanzapine-induced weight gain, without differences in the antipsychotic efficacy. However, an earlier meta-analysis [65] comparing the short-term effect of OLZ/SAM to olanzapine monotherapy on the weight and cardiometabolic parameters did not support these results, underlining the need for further research. In addition, a recent review on olanzapine/samidorphan concluded that there is no sufficient evidence on the use of the combination for the prevention of olanzapine-induced weight gain [66]. However, a more recent trial, including patients < 4 years since symptom onset with schizophrenia, schizophreniform disorder, or BD-I, showed that the OLZ/SAM treatment resulted in less weight gain versus olanzapine [67].

### 4.2. Lumateperone

Lumateperone is a butyrophenone atypical antipsychotic approved by the FDA in 2019 for the treatment of schizophrenia. It is a potent antagonist at 5-HT2A, a presynaptic partial agonist and a postsynaptic antagonist at the dopamine D2 receptors, and a dopamine receptor phosphoprotein modulator. It is also a D1 receptor-dependent indirect modulator of glutamatergic α-amino-3-hydroxy-5-methyl-4-isoxazolepropionic acid and N-methyl-D-aspartate (NMDA) GluN2B receptors and a blocker of the serotonin transporter (SERT) [68]. Thus, lumateperone simultaneously modulates 5-HT, DA, and glutamate neurotransmission. Its approval was based on the results from two randomized, double-blind placebo-controlled Phase II and III clinical trials, in which lumateperone showed efficacy as an antipsychotic agent in adults with schizophrenia in acute exacerbation [69,70]. Both studies aimed to assess the efficacy of lumateperone, using baseline PANSS differences. Lumateperone at 42 mg showed significant antipsychotic efficacy compared to the placebo. It also demonstrated a favorable safety profile not associated with extrapyramidal side-effects, weight gain, or metabolic changes [71].

Lumateperone is thus claimed to be an effective antipsychotic agent, appropriate as a first-line treatment for patients with schizophrenia, claiming a possible long-term positive impact on cognitive and negative symptoms [72]. Moreover, a systematic review including five clinical trials on the effect of lumateperone on body weight demonstrated minimum impact on weight gain [73], and in three phase II/III trials, the benefit-risk assessment of lumateperone was favorable, as measured by the number needed to treat (NNT), the number needed to harm (NNH), and the likelihood to be helped or harmed (LHH) [74].

### 4.3. Brilaroxazine (RP5063)

Brilaroxazine is an antipsychotic agent under development with a unique binding profile [75,76]. It acts as a partial agonist at the D2, D3, and D4 receptors and at the 5-HT1A and 5-HT2A receptors. It is also an antagonist at the 5-HT2B, 5-HT6, and 5-HT7 receptors and a full agonist at the nAChR α4β2 receptors. In addition, it modulates SERT with moderate affinity. It differs from other antipsychotics as it combines potent affinity and selectivity for the specific targets implicated in schizophrenia. As a result, brilaroxazine may be used as an antipsychotic with limited side-effects that are otherwise associated with commonly used antipsychotics.

In 2018, Cantillon and colleagues [77] published the results of a phase-I, double-blind, ascending-dose randomized clinical trial aiming to assess the safety of RP5063 in four cohorts, treated with 10, 20, 50, and 100 mg/day. In the single-dose, used in healthy volunteers, orthostatic hypotension, nausea, and dizziness were the most common side-effects, while in the multiple-dose, used in clinically stable patients with schizophrenia, akathisia and somnolence were the most frequently reported side-effects. Moreover, there were no significant changes in the lipid or prolactin levels, weight, and electrocardiograph (ECG) recordings. In the same study, significant improvements in positive symptoms were shown with brilaroxazine compared to the placebo in individuals with a baseline PANSS score higher than or equal to 50. Furthermore, promising preliminary findings of its efficacy on cognition were reported in patients under 50 mg of brilaroxazine.

A phase II, 4-week, double-blind, placebo- and aripiprazole-controlled clinical trial evaluated the safety and efficacy of brilaroxazine (15, 30, or 50 mg) in 234 patients with schizophrenia or schizoaffective disorder in acute episodes. It demonstrated that brilaroxazine at 15 mg and 50 mg significantly improved the PANSS total score compared to the placebo [75]. The most prevalent side-effects were insomnia and agitation. There were no metabolic deficits, abnormalities in ECG recordings, or risk of hypotension. Another interesting finding was improved cognitive and negative symptoms in the brilaroxazine groups. Currently, two phase III studies are ongoing: a randomized, double-blind, placebo-controlled, multicenter study to assess the safety and efficacy of brilaroxazine (RP5063) in subjects with acute exacerbation of schizophrenia; and an open-label, multicenter study to assess the safety and tolerability of brilaroxazine (RP5063) in subjects with stable schizophrenia (Table 1) [78].

### 4.4. Xanomeline/Trospium Combination (Kar-XT)

Xanomeline is a muscarinic ACh receptor agonist with reasonable selectivity for the M1 and M4 subtypes and an M5 receptor antagonist studied for the treatment of both Alzheimer’s disease and schizophrenia, particularly for the cognitive and negative symptoms [79,80,81,82]. In clinical trials, there was a high drop-out rate due to adverse reactions, commonly referred to as the cholinergic syndrome, encompassing mainly gastrointestinal symptoms such as sweating, nausea, vomiting, and diarrhea, but also excessive salivation, orthostasis, and syncope. Trospium is a peripheral muscarinic receptor antagonist. As it does not cross the blood-brain barrier (BBB), it does not act as an antagonist to the effects of xanomeline in the brain [83]. It has been shown to be effective in mitigating xanomeline-related cholinergic adverse effects. KarXT had an improved safety profile compared with xanomeline alone [84].

A recent phase II, 5-week, double-blind, randomized, placebo-controlled clinical trial evaluated the combination of xanomeline/trospium in 182 patients with schizophrenia in acute episodes [85]. The xanomeline/trospium combination resulted in a greater reduction in psychotic symptoms compared to the placebo. In a *post-hoc* analysis of the EMERGENT-1 study, KarXT was associated with a low overall adverse event burden [86]. The majority of the procholinergic and anticholinergic adverse events with KarXT were mild, occurred early during treatment, and were transient. There were no reports of an association between a higher incidence of extrapyramidal symptoms or weight gain.

The results of a phase III, randomized, double-blind, placebo-controlled clinical trial in 252 in-patients with schizophrenia have been presented as a poster [87,88]. Patients on the xanomeline/trospium combination demonstrated significant improvement by the end of week 5 (*p* < 0.0001). In the xanomeline/trospium group, the improvement in negative symptoms was slight but significant. Xanomeline/trospium was well-tolerated, with the most prevalent side-effects being cholinergic. There were no reports of any side-effects commonly associated with atypical antipsychotics. Furthermore, using the PANSS to assess the effectiveness, response was demonstrated as early as 2 weeks relative to the placebo. KarXT demonstrated improvements vs. the placebo in all five PANSS factors (positive symptoms, negative symptoms, disorganized thought, uncontrolled hostility, and anxiety/depression) [89]. There are five phase III trials currently ongoing, one of them not recruiting (Table 1).

### 4.5. Emraclidine (CVL-231)

Emraclidine is an M4-selective positive allosteric modulator (PAM) currently in development as a novel antipsychotic drug for the treatment of schizophrenia [51]. PAMs increase the activity of a single M4 receptor by binding to an allosteric binding site to increase the responsiveness of the receptor to acetylcholine by increasing the probability that the receptor will be activated by the neurotransmitter [49]. It selectively targets the activation of the M4 receptor in the brain, resulting in reduced dopaminergic activity without direct DA receptor antagonist activity. Thus, emraclidine could have an antipsychotic effect while minimizing the side-effects commonly linked to other antipsychotics.

The results of a two-part, phase Ib, randomized, placebo-controlled clinical trial were published in 2022 [90], and further trials are currently recruiting participants. In part A, the safety and tolerability profile of emraclidine was assessed in patients with stable schizophrenia, randomized in emraclidine 5–40 mg or placebo. In Part B, the safety and tolerability were evaluated in patients with schizophrenia in the acute phase, randomized in emraclidine 30 or 40 mg or placebo. Both doses of emraclidine showed a favorable safety and tolerability profile and significant improvement in psychotic symptoms, as reflected by a reduction in the PANSS total score (emraclidine 30 mg qd, *p* = 0.023; emraclidine 20 mg bid, *p* = 0.047). At present, three phase II clinical trials are ongoing to confirm the efficacy, safety, and tolerability of emraclidine (Table 1). Primary results are expected by the end of 2023.

### 4.6. Ulotaront (SEP363856)

Ulotaront is a novel compound with antipsychotic activity independent of D2 binding, confirmed in vivo in mice and by PET in non-human primates, and developed using SmartCube technology [91]. Ulotaront activates the TAAR1 receptors and ameliorates presynaptic DA dysfunction without D2 receptor binding. Ulotaront also seems to reverse glutamate hypofunction and activates the 5-HT1A receptors.

The results of a phase II, 4-week, randomized, double-blind, flexible-dose (50 mg/day or 75 mg/day) aimed to evaluate the efficacy and safety of SEP 363,856 in adults with schizophrenia in acute relapse were published in 2020 [92]. Ulotaront demonstrated significant superiority in improving the PANSS total score compared to the placebo (*p* = 0.001). Furthermore, it showed a favorable safety and tolerability profile. A six-month extension study followed up the recruited patients who completed the initial study to evaluate the safety and efficacy of long-term ulotaront treatment. Ulotaront use was associated with continued improvement in the PANSS total and BNSS total scores [93]. The most frequently reported side-effects were headache, insomnia, and anxiety, while treatment with ulotaront was associated with minimal risk of extrapyramidal symptoms, weight gain, and prolactinaemia. Based on the current data, ulotaront shows potential to be a first-in-class TAAR1 agonist for the treatment of schizophrenia with a safety and efficacy profile distinct from the current antipsychotics [94].

A population analysis [95] of the pharmacokinetics of ulotaront based on eight studies concluded that ulotaront shows a good absorption profile and exhibits dose-dependency at 10 to 100 mg. The median time for the maximum concentration and the median effective half-life were estimated to be 2.8 and 7 h, respectively [93]. Race, age, gender, drug formulation, or clinical status (healthy volunteer versus patient with schizophrenia) did not significantly impact the ulotaront pharmacokinetics. A recent systematic review [96] evaluated the available data on the safety, efficacy, and tolerability of ulotaront as a treatment for schizophrenia and suggested that it is a potentially effective antipsychotic agent with a good safety profile.

Ulotaront is in phase III clinical development for further evaluation of the efficacy, safety, and tolerability [97]. Two multicenter, randomized, double-blind, parallel-group, fixed-dosed studies (DIAMOND 1 and DIAMOND 2), evaluated the efficacy, safety, and tolerability of ulotaront (50 mg/day and 75 mg/day and 75 mg/day and 100 mg/day, respectively) versus the placebo over six weeks in 435 and 464, respectively, acutely psychotic adults with schizophrenia. All groups showed a reduction in the total PANSS score over time; however, neither of the ulotaront treatment groups were superior to the placebo on the primary endpoint of change from baseline in the PANSS total score at Week 6. In both studies, a large placebo effect was observed, which may have masked ulotaront’s therapeutic effect. The drug was generally safe and well-tolerated in both studies. In total, there are five phase III and one phase II/III study currently ongoing and recruiting (Table 1).

### 4.7. Pimavanserin (ACP-103)

Pimavanserin primarily functions as an inverse agonist and a partial inverse agonist at the 5-HT2A receptors. It spares the dopamine post-synaptic receptors, including the D2. Pimavanserin binds with high affinity to 5-HT2A, with lower affinity to 5-HT2C, and demonstrates negligible binding at the 5-HT2B, dopaminergic (D3), muscarinic (M5), and opioid (sigma 1) receptors [98]. Pimavanserin has a good safety profile. It is the first FDA-approved drug for the treatment of psychotic symptoms in patients with Parkinson’s disease [99]. The augmentation of antipsychotics with pimavanserin was shown to improve negative symptoms but failed to significantly reduce the total PANSS in two large, well-controlled double-blind studies [100] despite earlier promising add-on clinical studies [101]. A 2019 study found that pimavanserin improved the symptoms of schizophrenia and schizoaffective disorder in patients with refractory hallucinations and delusions who did not respond to clozapine or multiple antipsychotics [102]. As this was a study with ten participants only, further clinical trials are needed to confirm the effectiveness of pimavanserin vs. clozapine in patients with refractory schizophrenia or schizoaffective disorder.

ADVANCE-II, a phase II, 26-week, randomized, double-blind, placebo-controlled, multi-center, international study, assessed the efficacy and safety of pimavanserin in 403 stable patients with predominantly negative symptoms of schizophrenia while being on stable antipsychotic therapy [103]. Pimavanserin was well-tolerated, with a similar profile to the placebo regarding the side-effects. As for the efficacy, treatment with pimavanserin resulted in a statistically significant reduction in negative symptoms, but given the small effect size, the investigators suggested further studies for dose optimization.

ENHANCE-III, a phase III, 6-week, randomized, double-blind, placebo-controlled clinical trial, assessed pimavanserin as an adjunctive treatment in resistant-to-treatment out-patients with schizophrenia [104]. The study’s primary endpoint was a reduction in the PANSS total score, which was not met. However, pimavanserin demonstrated superiority regarding changes in the PANSS Negative Symptoms subscale and the Marder Negative Symptom Factor score compared to the placebo. Furthermore, pimavanserin showed a similar safety profile, the most common adverse events being headache, somnolence, and insomnia (Table 1).

### 4.8. Glutamatergic Modulation

#### 4.8.1. D-Amino Acid Oxidase (DAAO) Inhibitors

In the brain, D-amino acid oxidase (DAAO) is a peroxisomal flavoenzyme. Through oxidative deamination by DAAO, D-serine—the main co-agonist of synaptic N-methyl-D-aspartate (NMDA) receptors (NMDARs) [105]—is degraded into α-keto acids and ammonia. NMDAR hypofunction is implicated in the pathogenesis of schizophrenia. Patients with schizophrenia have lower D-serine levels in the peripheral blood and cerebrospinal fluid but higher DAAO expression and activity in the brain. Inhibiting DAAO activity and slowing D-serine degradation by using DAAO inhibitors to enhance NMDAR function has been employed in the treatment of schizophrenia.

It has been hypothesized that excessive DAAO activity may cause reduced D-serine levels, possibly contributing to the NMDAR hypofunction in schizophrenia [106]. DAAO is most active in the cerebellum, small intestine, liver, and kidney. Some experts believe that a DAAO inhibitor could be useful in treating schizophrenia by preventing the peripheral breakdown of D-serine, rather than through direct cortical activity.

##### Sodium Benzoate (SND14, NaBen^®^), SyneuRx International, New Taipei City, Taiwan

Sodium benzoate (SyneuRx International Corp, Taiwan), an accessible food preservative, is a D-amino acid oxidase (DAAO) inhibitor, initially shown to increase synaptic D-serine levels [107]. However, it does not appear to impact D-serine levels in vivo, and the exact mechanism of action remains unclear [108].

In 2013, Lane and colleagues presented the results of a 6-week, randomized, double-blind, placebo-controlled trial that examined the efficacy and safety of adjunctive sodium benzoate at 1 g/day, in 52 patients with stable schizophrenia for at least three months [109]. The study’s primary endpoint was a reduction in the PANSS total score. The sodium benzoate group showed significant improvement in the PANSS total and subscales scores (*p* < 0.001 for all; ES  =  1.16–1.69). Additionally, a neurocognitive assessment revealed the superiority of the sodium benzoate treatment compared to the placebo, especially in terms of processing speed and visual memory. Sodium benzoate exhibited a good safety profile without significant adverse effects.

In a 6-week [110], double-blind, clinical trial, 60 inpatients with schizophrenia under clozapine treatment were randomized to 1 g/day sodium benzoate, 2 g/day sodium benzoate, or placebo, to assess the efficacy and safety of sodium benzoate as an add-on therapy. In this study, both groups of sodium benzoate produced a significant improvement in negative symptoms compared to the placebo, while at 2 g/day, sodium benzoate showed superiority in the PANSS total score, with a favorable safety profile. There was no improvement in cognitive function in either dose, however.

In contrast, a 12-week, double-masked, placebo-controlled, randomized clinical trial from Australia [111] examining the efficacy of sodium benzoate in 100 patients with early psychosis (age range 15 to 45) failed to prove the superiority of adjunctive 500 mg sodium benzoate bd compared to the placebo in any of the scales used for the evaluation.

A recent meta-analysis from India [112] evaluated the efficacy and safety of add-on sodium benzoate for the treatment of schizophrenia. They showed that add-on therapy with sodium benzoate improved the positive symptoms of schizophrenia but there was no evidence of improving the negative symptoms, general psychopathology, cognitive function, social, occupational, psychological functioning, and quality of life. Indeed, the occurrence of significantly higher extrapyramidal symptoms raised concerns about the safety of sodium benzoate. Hence, considering the existing evidence and its limited efficacy and adverse effect profile, sodium benzoate may not be considered an add-on therapy in schizophrenia.

As the hypofunction of the NMDA receptor is implicated in the pathophysiology, particularly cognitive impairment, of schizophrenia, in a randomized, double-blind, placebo-controlled trial in patients with chronic schizophrenia, add-on sodium benzoate was given in combination with sarcosine, a glycine transporter I (GlyT-1) inhibitor, to enhance NMDA receptor-mediated neurotransmission (see below). The investigators concluded that the combination of these two NMDA-enhancing agents, but not sarcosine alone, can improve cognitive function, despite the non-improvement in the clinical symptoms of patients with chronic schizophrenia [113].

To further clarify the therapeutic potential of sodium benzoate in schizophrenia, there are two ongoing phase II/III clinical trials on the safety and efficacy of NaBen^®^ as an add-on treatment for schizophrenia, one in adolescents and one in adults (Table 1).

##### Luvadaxistat (TAK-831), Neurocrine Biosciences, San Diego, CA, USA

In a phase II study, add-on therapy with another DAAO inhibitor (TAK-831/luvadaxistat; Neurocrine Biosciences Inc., San Diego, CA, USA) failed to address the negative symptoms of schizophrenia but resulted in cognitive symptom enhancement [114], and a phase II study in patients with cognitive impairment associated with schizophrenia (NCT05182476, Table 1) is underway.

Currently, the most reliable way to assess NMDAR-related function in humans is through mismatch negativity (MMN), which relies on the NMDAR’s dual voltage- and ligand-sensitivity. MMN is an event-related brain potential that is sensitive to stimulus deviation from a repetitive pattern, thought to reflect the activity of sensory memory, with, at most, moderate influences of higher-level cognitive processes, such as attention [115]. MMN is reported to be reduced in patients with chronic schizophrenia, presenting a biomarker in schizophrenia. As such, it was used in a randomized, placebo-controlled, double-blind, two-period crossover phase IIa study assessing luvadaxistat, a DAAO inhibitor under development for the treatment of cognitive impairment in schizophrenia. A nominally significant improvement in MMN was observed, suggesting that luvadaxistat 50 mg (vs. 500 mg) can improve this illness-related circuitry biomarker at doses associated with partial DAAO inhibition [116].

##### NMDA Function Enhancers

An early meta-analysis including 29 trials and a total of 1253 cases aiming to evaluate the potential of NMDA receptor modulators as adjunctive therapy for schizophrenia found D-serine, N-acetyl-cysteine (NAC), and sarcosine to have therapeutic benefit in the treatment of total and negative symptoms as adjuncts to non-clozapine antipsychotics. However, while glycine improved the positive and total symptoms as an adjuvant to non-clozapine antipsychotics, it worsened them when added to clozapine [117]. A more recent meta-analysis [118] showed that NMDAR modulator add-on treatments to antipsychotic therapy improved several schizophrenia symptoms, mainly negative symptoms. In addition, they had a satisfactory safety profile and side effects. Out of the seven glutamatergic agents studied, glycine, D-serine, and sarcosine showed better treatment profiles than the other agents. Indeed, adding NMDAR co-agonists reduced the schizophrenia symptoms compared to antipsychotic treatments as usual. However, the effectiveness of the augmentation with NMDAR modulators was only observed in patients treated with antipsychotics other than clozapine.

In addition to its powerful role as an inhibitory neurotransmitter through its binding to the Glycine Receptor (GlyR) in the spinal cord and the brainstem, glycine, a non-essential amino acid, is an important co-agonist of the NMDAR in excitatory glutamatergic transmission [119]. Various methods have been used to enhance NMDA function, such as administering glycine or other NMDAR co-agonists like D-serine or preventing their breakdown or reuptake. However, studies on glycine agonists have yielded inconsistent results, and no FDA-approved treatments have been developed yet [120].

In the synaptic cleft, glycine’s reuptake is regulated by glycine transporters, namely type I (GlyT1) and type II (GlyT2). GlyT1 is considered a highly potential target for psychosis therapy. A phase II study conducted on bitopertin, a selective and non-competitive GlyT1 inhibitor, demonstrated benefits for negative symptoms in a per-protocol population [121]. However, a subsequent phase III study on patients with predominant negative symptoms failed to show significant benefits [122]. The failure of bitopertin may be related to its U-shape dose-response curve of partial receptor occupancy that increases the glycine levels merely at extra-synaptic sites and is not sufficient for processes that induce long-term potentiation to correct a certain degree of NMDA receptor hypofunction [123].

##### Iclepertin (BI 425809), Boehringer Ingelheim, Ingelheim, Germany

Iclepertin (Boehringer Ingelheim, Ingelheim, Germany) is a potent and selective GlyT1 that showed significant (d  =  0.34) improvements in cognition over 12 weeks of treatment in patients with schizophrenia during a phase II study [124]. Another phase II study (NCT03859973) evaluating add-on iclepertin to the current antipsychotic therapy and computer-based training for cognitive symptoms of schizophrenia was recently completed. Four phase III studies of add-on iclepertin in schizophrenia are ongoing (NCT04846868, NCT04846881, NCT04860830, NCT05211947, Table 1).

### 4.9. Other Compounds

#### 4.9.1. Voltage-Gated Sodium Channel Blockers

Voltage-gated sodium channel blockers, in particular the antiepileptic drugs carbamazepine, oxcarbazepine, phenytoin, and lamotrigine, increase the motor threshold, and the corticospinal system becomes less excitable [125]. However, they have been extensively investigated in the treatment of schizophrenia with no promising results. Interestingly, the antipsychotic aripiprazole, a D2 partial agonist, is a potent voltage-gated sodium channel blocker. It is also an agonist at the D3 and 5-HT1A receptors and an antagonist at the 5-HT2A receptors. Owing to its pharmacology, aripiprazole was shown to reduce antipsychotic-induced hyperprolactinemia, produce less kinetic disorders, and cause fewer cardiovascular adverse reactions compared to other antipsychotics, in addition to possessing possible anticancer and neuroprotective properties [126].

Evenamide (NW-3509, Newron Pharmaceuticals, Bresso, Italy) is a voltage-gated sodium channel blocker and inhibits the synaptic release of glutamate. It thus reduces hyperexcitability in the PFC and the hippocampus. Unlike other antipsychotics, it does not interact with other neurotransmitter systems (dopaminergic, noradrenergic, serotonergic, and histaminergic), acts preferentially on sodium channels, and modulates glutamate release. It was shown to reverse the NMDR antagonist ketamine- and phencyclidine-induced reduction in prepulse inhibition [127] and has thus been proposed as a promising compound for investigation in the treatment of schizophrenia.

Indeed, the results from a pilot, 6-week, randomized, open-label, rater-blinded study with a 46-week extension indicate very good tolerability with an exceptional, clinically important, increase in the efficacy of evenamide (7.5, 15, and 30 mg bid) as an add-on treatment to antipsychotics in treatment-resistant schizophrenia patients [128].

#### 4.9.2. L-Carnosine

L-Carnosine is an antioxidant reported to improve negative and cognitive symptoms in schizophrenia [129]. A randomized, double-blind, placebo-controlled study examined the effectiveness of adjuvant L-Carnosine therapy in schizophrenia. A total of 100 eligible patients with predominant negative symptoms were randomly assigned to receive either a fixed dose of 400 mg L-Carnosine or an identical placebo for 3 months and increased to 800 mg from the 13th week until the study end. The attention scores were significantly different in patients receiving 800 mg L-Carnosine (*p* = 0.023), but there were no significant differences in the negative symptoms between groups. In conclusion, it has been suggested that a dosage of 800 mg of L-Carnosine was promising for improving executive functions in schizophrenia [130]. 

## 5. Conclusions

The treatment of schizophrenia remains a significant challenge in clinical practice, despite the enormous progress in psychopharmacological research. The complexity of its aetiopathogenesis leads to high variability in the medication response across patients with schizophrenia and schizoaffective disorder. While several psychotropic drugs are currently available, one out of three patients remains a non-responder [131], the effect on negative and cognitive symptoms is limited, and there are still debilitating residual symptoms, such as extrapyramidal side-effects and other drug-related adverse effects. As a result, it is crucial to strengthen the clinical research on novel antipsychotic agents.

Novel compounds in the pipeline for use in schizophrenia appear to work beyond the dopamine system (Table 2).

Future drug discovery should focus on the optimal implementation of the available treatments, emphasizing improvements in medication adherence due to improved side-effect profiles, and hence adherence. Long-acting or dual preparations are a good example of such strategies. Most importantly, based on novel mechanisms of action, prospective medications should address the debilitating negative and cognitive symptoms, in addition to treating the positive symptoms, whilst avoiding the burden of post-synaptic DA blockade-induced side-effects.

## Figures and Tables

**Table 1 brainsci-13-01193-t001:** Emerging Treatments for Schizophrenia, www.clinicaltrials.gov, accessed on 31 July 2023.

Investigational Agent	Mechanism of Action	Official Study Title	ClinicalTrials.gov ID	Ongoing Phase II/III Trials in Schizophrenia
RP-5063 (Brilaroxazine) [Reviva Pharmaceuticals, Cupertino, CA, USA]	High affinity for serotonin (5-HT) 1A/2A/2B/7 and dopamine (D) 2/3/4 and moderate affinity for D1, serotonin transporter (SERT), and nicotinic acetylcholine receptor, α4β2	Phase III, Randomized, 28 Days, Double-blind, Placebo-controlled, Multicenter Study to Assess the Safety and Efficacy of Brilaroxazine (RP5063) in Subjects With Schizophrenia, Followed by a 52-Week Open-label Extension	NCT05184335	Phase III, Ongoing—Recruiting
		Phase III, Randomized, 28 Days, Double-blind, Placebo-controlled, Multicenter Study to Assess the Safety and Efficacy of Brilaroxazine (RP5063) in Subjects With Schizophrenia, Followed by a 52-Week Open-label Extension	NCT05184335	Phase III, Ongoing—Recruiting
KarXT (Xanomeline and Trospium Chloride) [Karuna Therapeutics, Boston, MA, USA]	A direct muscarinic cholinergic receptor agonist preferentially acting at M1 and M4 receptors	A Phase III, Multicenter, Two-part Study With a 5-week Double-blind Part (Randomized, Parallel-group, Placebo-controlled) Followed by a 12-week Open-label Extension Part, to Evaluate the Efficacy and Safety of KarXT in Acutely Psychotic Hospitalized Chinese Adult Subjects With DSM-5 Schizophrenia	NCT05919823	Phase III, Ongoing—Recruiting
		An Open-label Extension Study to Assess the Long-term Safety and Tolerability of Adjunctive KarXT in Subjects With Inadequately Controlled Symptoms of Schizophrenia	NCT05304767	Phase III, Ongoing—Recruiting
		An Open-label Study to Assess the Long-term Safety, Tolerability, and Efficacy of KarXT in De Novo Subjects With DSM-5 Schizophrenia	NCT04820309	Phase III, Ongoing—Recruiting
		An Open-label Extension Study to Assess the Long-term Safety, Tolerability, and Efficacy of KarXT in Subjects With DSM-5 Schizophrenia	NCT04659174	Phase III, Ongoing—Not Recruiting
		A Phase III, Randomized, Double-blind, Placebo-controlled Study to Evaluate the Safety and Efficacy of Adjunctive KarXT in Subjects With Inadequately Controlled Symptoms of Schizophrenia	NCT05145413	Phase III, Ongoing—Recruiting
CVL-231 (Emraclidine) [Cerevel Therapeutics, Cambridge, MA, USA]	A muscarinic M4 receptor positive allosteric modulator	A Phase II, Randomized, Double-blind, Placebo-controlled Trial to Evaluate the Efficacy, Safety, and Tolerability of Two Fixed Doses (15 mg and 30 mg QD) of CVL-231 (Emraclidine) in Participants With Schizophrenia Experiencing an Acute Exacerbation of Psychosis	NCT05227703	Phase II, Ongoing—Recruiting
		A Phase II, Randomized, Double-blind, Placebo-controlled Trial to Evaluate the Efficacy, Safety, and Tolerability of Two Fixed Doses (10 mg and 30 mg QD) of CVL-231 (Emraclidine) in Participants With Schizophrenia Experiencing an Acute Exacerbation of Psychosis	NCT05227690	Phase II, Ongoing—Recruiting
		A 52-week, Phase II, Open-label Trial to Evaluate the Long-term Safety and Tolerability of CVL-231 (Emraclidine) in Adult Participants With Schizophrenia	NCT05443724	Phase II, Ongoing—Recruiting
SEP-363856 (Ulotaront) [Sumitomo Pharma, Osaka, Japan]	A trace-amine associated receptor 1 (TAAR1) agonist with 5-HT1A receptor agonist activity	A Randomized, Double-blind, Parallel-group, Placebo-controlled, Fixed-dose, Multicenter Study to Evaluate the Efficacy and Safety of SEP-363856 in Acutely Psychotic Subjects With Schizophrenia	NCT04092686	Phase III, Ongoing—Recruiting
		A Randomized, Double-blind, Parallel-group, Placebo-controlled, Fixed-dose, Multicenter Study to Evaluate the Efficacy and Safety of SEP-363856 in Acutely Psychotic Subjects With Schizophrenia	NCT04072354	Phase III, Ongoing—Recruiting
		A Randomized, Double-blind, Parallel-group, Placebo-Controlled, Fixed-dose, Multicenter Study to Evaluate the Efficacy and Safety of SEP 363,856 in Acutely Psychotic Patients With Schizophrenia, Followed by an Open-label Extension Phase	NCT04825860	Phase II/III, Ongoing—Recruiting
		An Open-label Extension Study to Assess the Safety and Tolerability of SEP-363856 in Subjects With Schizophrenia	NCT04109950	Phase III, Ongoing—Recruiting
		A 52-week, Open-label Study to Evaluate the Long-term Safety and Tolerability of SEP-363856 in Patients With Schizophrenia in Japan	NCT05359081	Phase III, Ongoing—Recruiting
		An 8-Week, Open-Label Study Evaluating the Effectiveness, Safety and Tolerability of SEP-363856 in Subjects With Schizophrenia Switched From Typical or Atypical Antipsychotic Agents	NCT05628103	Phase III, Ongoing—Recruiting
ACP-103 (Pimavanserin) [ACADIA Pharmaceuticals, San Diego, CA, USA]	A selective inverse agonist of the 5-HT2A receptor	A Phase III, Randomized, Double-Blind, Placebo-Controlled Study to Evaluate the Efficacy and Safety of Pimavanserin as Adjunctive Treatment for the Negative Symptoms of Schizophrenia	NCT04531982	Phase III, Ongoing—Recruiting
		A 52-Week, Open-Label, Extension Study of Pimavanserin for the Adjunctive Treatment of Schizophrenia	NCT03121586	Phase III, Ongoing—Recruiting
SND-14 (Sodium Benzoate, NaBen^®^) [SyneuRx International, New Taipei City, Taiwan]	A D-amino acid oxidase (DAAO) inhibitor	An Adaptive, Phase IIb/III, Double-Blind, Randomized, Placebo-Controlled, Multi-Center Study of the Safety and Efficacy OF NaBen^®^, A D-Amino Acid Oxidase Inhibitor, as an Add-on Treatment for Schizophrenia in Adolescents	NCT01908192	Phase II/III, Ongoing—Recruiting
		An Adaptive, Phase IIb/III, Multi-center, Prospective, Randomized, Double-Blind Placebo-controlled Study of the Safety and Efficacy of NaBen^®^ (DAAO Inhibitor), as an Add-on Treatment for Schizophrenia in Adults	NCT02261519	Phase II/III, Ongoing—Recruiting
TAK-831 (Luvadaxistat) [Neurocrine Biosciences, San Diego, CA, USA]	A D-amino acid oxidase (DAAO) inhibitor	A Randomized, Double-Blind, Placebo-Controlled, Parallel-Group Study to Evaluate the Efficacy, Safety, and Tolerability of Luvadaxistat in Subjects With Cognitive Impairment Associated With Schizophrenia, Followed by Open-Label Treatment	NCT05182476	Phase II, Ongoing—Recruiting
BI-425809 (Iclepertin) [Boheringer Ingelheim, Ingelheim, Germany]	A glycine transporter-1 (GlyT1) inhibitor	A Phase III Randomized, Double-blind, Placebo-controlled Parallel Group Trial to Examine the Efficacy and Safety of Iclepertin Once Daily Over 26 Week Treatment Period in Patients With Schizophrenia (CONNEX-1)	NCT04846868	Phase III, Ongoing—Recruiting
		A Phase III Randomized, Double-blind, Placebo-controlled, Parallel Group Trial to Examine the Efficacy and Safety of Iclepertin Once Daily Over 26 Week Treatment Period in Patients With Schizophrenia (CONNEX-2)	NCT04846881	Phase III, Ongoing—Recruiting
		An Open Label, Single Arm, Extension Trial to Examine Long-term Safety of Iclepertin Once Daily in Patients With Schizophrenia Who Have Completed Previous Iclepertin Phase III Trials (CONNEX-X)	NCT05211947	Phase III, Ongoing—Recruiting
		A Phase III Randomized, Double-blind, Placebo-controlled Parallel Group Trial to Examine the Efficacy and Safety of Iclepertin Once Daily Over 26 Week Treatment Period in Patients With Schizophrenia (CONNEX-3)	NCT04860830	Phase III, Ongoing—Recruiting

**Table 2 brainsci-13-01193-t002:** The pharmacological binding profile of novel antipsychotics **.

	Target System	Dopamine	Serotonin	Noradrenaline	Histamine	Acetylcholine	TAAR	Glutamate
Compound								
Olanzapine/ Samidorphan		D4 (−) D2 (−) D3 (−) D1 (−)	5HT2A (−) 5HT6 (−) 5HT2B (−) 5HT2C (−) 5HT3 (−) 5HT7 (−) 5HT1B (−) 5HT1D (−)	a2C (−) α1A (−) α2B (−) α2A (−) α1B (−)	H1 (−)	M1 (−) M3 (−) M2 (−) M4 (−)		
Lumateperone		D2 (postsynaptic) (−) D2 (presynaptic) (+) D1 (−)	5HT2A (−) 5HT2c (−) SERT (−)	α1 (−)				
Xanomeline/ Trospium			5HT2B (−) 5HT2C (−) 5HT1B (+) 5HT1A (+) 5HT2A (−)			M1 (+) M3 (+) M2 (+) M4 (+) M5 (−)		
Brilaroxazine		D2 (+) (*) D3 (+) (*) D4 (+) (*)	5HT1A (+)(*) 5HT2A (+)(*) 5HT2B (−) 5HT6 (−) 5HT7 (−)			nAChR α4β2 (+)		
Ulotaront			5HT1A (+) 5HT1D (+) 5HT1B (+) 5HT7 (+)				TAAR1 (+)	
Emraclidine						M4 (*)		
Pimavanserin			5HT2A (−) 5HT2C (−)					
Sodium Benzoate								DAAO (i)
Luvadaxistat								DAAO (i)
Iclepertin								GlyT1 (−)

Abbreviations: (+) agonist; (−) antagonist; (+) (*) partial agonist; (*) positive allosteric modulator; (i) inhibitor; ** Receptors are referred to in order of binding strength.

## Data Availability

Not applicable.

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
