# Peer review of "Novel Compounds in the Treatment of Schizophrenia—A Selective Review"

_brainsci, 2023, doi:10.3390/brainsci13081193_

Round 1

Reviewer 1 Report

This review discusses the current hypothesis leading to schizophrenia and novel compounds recently approved or in the pipeline.  This is a useful review as there are currently a number of new promising compounds in development.

I have a few minor suggestions to improve the manuscript.  

1)  Under the serotonin section, there are two paragraphs that focus on the glutamate system.  This could possibly be reorganized since there is a section on glutamate.

2) The last paragraph on Pimavanserine and clozapine non responders should be taken out.  This is more of a case series and the rest of the review focuses on clinical trials.  There were a number of limitations to that study and it needs replication in a clinical trial.  

3) The authors discuss glutamate at length but do not describe any of the compounds in development related to glutamate such as NMDA antagonists.  They do not discuss D-serine, as well as other compounds such as DAAO inhibitors, and voltage gated sodium channel blockers.  If they are not including these, they should lay out in the paper why they are choosing some compounds in development and not others. It does not appear that this is an exhaustive review of all the compounds in development.  What are the criteria for inclusion?

Author Response

1) Under the serotonin section, there are two paragraphs that focus on the glutamate system. This could possibly be reorganized since there is a section on glutamate.

This paragraph is now substantially changed according to the reviewer’s comment, thank you.

2) The last paragraph on pimavanserin and clozapine non-responders should be taken out. This is more of a case series and the rest of the review focuses on clinical trials. There were a number of limitations to that study and it needs replication in a clinical trial.

We have now shortened this paragraph and highlighted the need for further trials.

3) The authors discuss glutamate at length but do not describe any of the compounds in development related to glutamate such as NMDA antagonists. They do not discuss D-serine, as well as other compounds such as DAAO inhibitors, and voltage-gated sodium channel blockers. If they are not including these, they should lay out in the paper why they are choosing some compounds in development and not others. It does not appear that this is an exhaustive review of all the compounds in development.

Thank you for this comment. We have updated our review accordingly.

What are the criteria for inclusion? We have focused on the MEDLINE search, as detailed in the ‘new’ Methods section, which has now been added. Based on this comment, we broadened our search to include more compounds for completeness. However, our review still focusses on promising compounds in the pipeline.

Reviewer 2 Report

Schizophrenia is a multifactorial, highly complex behavioral and cognitive disorder caused by disruptions of neurotransmitters in the brain, consequently affecting its functioning. The disorder is known to affect approximately 1% of the adult population worldwide.This narrative review of the literature aims to present recently approved and newly discovered pharmacological substances for the treatment of schizophrenia as well as their suggested mechanism of action.In my opinion, this manuscript is worth publishing.

 Minor editing of English language required

Author Response

Comments on the Quality of English Language

Minor editing of English language required.

Thank you. We have edited the language and corrected all grammatical issues found.

Reviewer 3 Report

This manuscript entitled “Novel Compounds for the Treatment of Schizophrenia – A Narrative Review” is well written and informative. I only have minor suggestions.

There are several sections in which the authors need to add references. To provide some examples:

“serotoninergic” psychosis line 104

Paragraph from lines109 to 113

After sentence from line 200 to 202

At the end of lines 204 and then 205

At the end of sentence 414

The authors also state that one out of three patients remain non-responders line 460 needs a reference

Other minor suggestions.

Section on Olanzapine/Samidorphan, line 279 starts with “Interestingly” and in my opinion is should say “As expected”.

On this section the authors should include the assessment of Monahan et al, Annals of Pharmacotherapy, 2022.

Lines 366 and 367, authors refer to cholinergic side effects. It would be important to list them.

On paragraph lines 83 to 87, recent work that support their proposition has been recently published see Bellon et al, Molecular Psychiatry, 2022.

Author Response

Reviewer 3

This manuscript entitled “Novel Compounds for the Treatment of Schizophrenia – A Narrative Review” is well written and informative. I only have minor suggestions.

There are several sections in which the authors need to add references. To provide some examples:

serotoninergic” psychosis line 104

We added the reference.

Paragraph from lines109 to 113

We added the reference.

After sentence from line 200 to 202

We added the reference.

At the end of lines 204 and then 205

We added the reference.

At the end of sentence 414

We added the reference.

The authors also state that one out of three patients remain non-responders line 460 needs a reference

We added the reference.

Other minor suggestions.

Section on Olanzapine/Samidorphan, line 279 starts with “Interestingly” and in my opinion is should say “As expected”.

Thank you. The suggested phrase was used.

On this section the authors should include the assessment of Monahan et al, Annals of Pharmacotherapy, 2022.

Thank you. The suggested study was added.

Lines 366 and 367, authors refer to cholinergic side effects. It would be important to list them.

We added this sentence: “Due to gastrointestinal side-effects, there was a high drop-out rate in clinical trials due to adverse reactions, commonly referred to as the cholinergic syndrome, encompassing symptoms such as sweating, nausea, vomiting, diarrhea, excessive salivation, orthostasis, and syncope.” Thank you.

On paragraph lines 83 to 87, recent work that support their proposition has been recently published see Bellon et al, Molecular Psychiatry, 2022.

The suggested study was added.